# Boon and Bane of DNA Double-Strand Breaks

**DOI:** 10.3390/ijms22105171

**Published:** 2021-05-13

**Authors:** Ingo Schubert

**Affiliations:** Leibniz Institute of Plant Genetics and Crop Plant Research (IPK), OT Gatersleben, D-06466 Seeland, Germany; schubert@ipk-gatersleben.de

**Keywords:** DNA double-strand break (DSB) repair, meiotic crossover, chromosome rearrangements, differentiation, speciation, evolution, gene technology

## Abstract

DNA double-strand breaks (DSBs), interrupting the genetic information, are elicited by various environmental and endogenous factors. They bear the risk of cell lethality and, if mis-repaired, of deleterious mutation. This negative impact is contrasted by several evolutionary achievements for DSB processing that help maintaining stable inheritance (correct repair, meiotic cross-over) and even drive adaptation (immunoglobulin gene recombination), differentiation (chromatin elimination) and speciation by creating new genetic diversity via DSB mis-repair. Targeted DSBs play a role in genome editing for research, breeding and therapy purposes. Here, I survey possible causes, biological effects and evolutionary consequences of DSBs, mainly for students and outsiders.

## 1. Background

The heritable information of all living beings on Earth (except for some RNA, or single-stranded DNA, viruses) is encoded, stored and propagated through the base sequence of complementary double-stranded deoxyribonucleic acid molecules. The double-stranded DNA forms circular (in most prokaryotes, plastids and mitochondria) or linear molecules (in eukaryotic nuclei). Eukaryotic chromosomes, the carriers of linear nuclear DNA, are organized by histones and other nuclear proteins in hierarchical structures which differ in compactness during differentiation and along the cell cycle.

For correct reading, replication and segregation of double-stranded DNA molecules, local interruption of both strands (double-strand break = DSB) is the most critical lesion. DSBs may occur via mechanic shearing, ionizing irradiation, some chemical mutagens such as bleomycin and reactive oxygen species (ROS) or enzymatically, mediated by endonucleases. Endonucleases cut double strands directly, or opposite single-stranded break positions and excision repair gaps or at stalled replication forks (e.g., [1]). Adjacent single-strand breaks can complement each other to DSBs [2]. DSBs and their processing may occur in all cells of an organism, and the products can be transferred via germ line (or plant apical meristems) into the next generation.

## 2. Unrepaired DSBs Can Be Lethal for Dividing Cells

Unrepaired DSBs may have fatal consequences for cells harboring the broken molecules. In the case of circular DNA molecules, open break ends provide a substrate for exonucleases, degrading the molecules from their ends (for review: [3]). In addition, complete regular replication is no longer possible, because all known DNA polymerases work only in 5′ to 3′-direction and require an RNA primer to start polymerization at a free 3′-hydroxyl group. The primer is later replaced by DNA. Therefore, free break ends would be shortened with each round of replication.

In the case of linear nuclear DNA molecules, in addition to what may happen to broken circular ones, correct segregation of broken fragments into daughter nuclei is usually not possible. Rare exceptions are fragments of holocentric chromosomes to which spindle fibers attach along the entire length (for review: [4]) and occasional ‘hitchhiking’ of acentric fragments together with undamaged chromosomes [5]. An acentric fragment of broken monocentric chromosomes (the break product from distal the centromere at which spindle fibers attach to pull the sister chromatids to opposite cell poles) usually segregates at random and eventually gets lost. The break end at the centric fragment is also prone to degradation and to ‘illegitimate recombination’, as long as no stabilizing telomere structure is added. Illegitimate recombination means either interaction with other break ends, yielding an aberrant linkage, or invasion into allelic or ectopic homologous double-stranded regions for elongating via replication (see below). If the break end is elongated up to the end of the template molecule, the process is called ‘break-induced replication’ (BIR) which is experimentally proven for yeast [6].

## 3. DSBs Can Be Repaired by Diverse Mechanisms

Because of the negative impact of DSBs on cell survival, several mechanisms of their repair were adopted during evolution already in prokaryotes and extended in eukaryotes. These mechanisms are characterized mainly by a different degree of homology requirements at break ends (Figure 1) and a particular enzyme equipment. They comprise direct ligation of ‘clean’ break ends (non-homologous end-joining = NHEJ), or microhomology-mediated (= alternative) end-joining (MMEJ) which require a few complementary bases at single-stranded overhangs of the break ends to align prior to ligation (for review: [7,8]). Another mechanism is ‘single-strand annealing’ (SSA) which requires longer single-stranded overhangs and larger regions of homology to pair before ligation (for instance the long terminal repeats = LTRs of LTR-retrotransposons) (for review: [9]). Other types of DSB repair, the variants of homology-directed recombination repair (HDR), include invasion of single-stranded overhanging break ends into (partially) homologous sequences, and their elongation over the broken region along the template sequence before resolution of the recombination structure (Holiday junction(s)) in a different manner (Figure 1) (for review: [10]).

## 4. DSB Repair Generates Diverse Phenomena

Depending on the way of resolution of the recombination structure, the sequence of the repair template (donor) can be transferred to the originally damaged recipient molecule via single-strand or via double-strand exchange (Figure 1). An elongated single-stranded end serves, after ligation to the opposite break end, as a template for gap-filling in the complementary strand (synthesis-dependent strand annealing = SDSA = gene conversion). A double-strand exchange is accompanied by a switch of flanking sequences between the donor and the recipient region. Depending on the chromosomal region which serves as a repair template, the outcome of a double-strand exchange differs (Figure 2). If the template is the identical region of the sister chromatid, a sister chromatid exchange (SCE) is the consequence (Figure 2A). If it is the allelic region of the homologous chromosome, a homologous translocation (or a cross-over, when occurring during meiotic prophase I) occurs (Figure 2B). Direct end-joining (NHEJ), SCE and SDSA with the identical allele usually have no genetic consequences and are therefore considered to be correct DSB repair. In case the template was an ectopic (partially) homologous region in *cis* or in *trans,* the outcome of recombination repair is either an intra- or an interchromosomal reciprocal translocation. Translocations can occur either via NHEJ between two breaks or via resolution of recombination structures during HDR. Interchromosomal translocations are symmetric and yield monocentric products, when proximal and distal break ends are ligated crosswise with each other (Figure 2B,C). They are asymmetric, if ligation involves proximal with proximal, and distal with distal ends. An asymmetric interchromosomal exchange yields a dicentric and an acentric fragment (Figure 2D). Both products of an asymmetric exchange are unstable during nuclear divisions. Dicentrics can cause bridges during nuclear divisions if the centromeres are distant enough that sister chromatids can twist between them (Figure 3B). Disrupted bridges may result in monocentric chromosomes with deletions or duplications, when after the first nuclear division telomeres are added (Figure 3C left box). If, after replication, the break ends of the sister chromatids are joined, a new bridge forms in the next division (Figure 3C right box) resulting in complex rearrangements (‘breakage-fusion-bridge cycles’ according to [12]). Acentrics are usually lost during subsequent nuclear divisions.

Intrachromosomal translocation leads to an inversion (Figure 2E) if the proximal ends and the distal ends of two breaks are crosswise ligated. When the proximal end of the first break is ligated with the distal end of the second break, and the distal end of the first with the proximal end of the second break, the result is a ring chromosome and a linear fragment (Figure 2F). If the ring is part of one chromosome arm, it behaves as an acentric. If the ring harbors the centromere, and after replication experiences odd numbers of sister chromatid exchanges (even numbers compensate each other) resulting in a double-sized dicentric ring, it gets disrupted and lost during later divisions (Figure 2G). 

DSBs and their repair can happen in all cell cycle stages. If the DSB and its mis-repair occur in G1, the entity of exchange is the entire unreplicated chromosome. After replication, both chromatids of the translocated chromosomes are identical. If breakage and repair occur after replication (during the S or G2 phase), only one chromatid of each involved chromosome enters nuclear division in the translocated state. Consequently, one daughter cell receives the wild-type and the other the translocated chromosomes when the segregation is balanced. If the segregation is unbalanced, the genetic constitution of the daughter cells (if viable) gives the impression of non-reciprocal translocation (Figure 4). The results are either loss of heterozygosity (if the involved chromosomes were homologs), or a complementary duplication/deletion between daughter nuclei (if the involved chromosomes were heterologs). However, non-reciprocal translocations have never been observed in statu nascendi, and unbalanced segregation (or alternatively BIR in yeast) is the more likely interpretation of the observed phenomenon.

## 5. Deleterious and Beneficial Consequences of DSB Repair

From a conservative point of view, DSBs, if unrepaired or incorrectly repaired, change the genetic constitution of a cell (or an organism, if transferred to gametes) and are to be considered as deleterious because they are lethal or counteract the maintenance of genetic information, which so far had proven as successful during evolution. However, some DSBs and their processing are important for developmentally regulated genome rearrangements and/or turn out to be drivers of evolution as exemplified in the following. All phenomena described in the following are experimentally proven, and are valid for all eukaryotes, if not stated otherwise (see 8, 9 and 11). 

DSBs which are programmed during meiotic prophase I are repaired in their majority without genetic consequences; a minority, via **cross-over** with the homologous allele, result in a new combination of maternal and paternal alleles. Cross-overs keep maternal and paternal homologous chromosomes together until reductional anaphase I, and thus enable correct segregation of parental chromosomes into gametes. The new combination of parental alleles, if beneficial for survival (and propagation) of its carriers, will be positively selected in the next generations.Chromosome **translocations** in the heterozygous state potentially reduce the fertility of carriers (due to the risk of lethality after unbalanced segregation) (Figure 4). Heterozygous **inversions**, if the corresponding regions engage in crossing over, will yield duplications and deletions, which are mostly lethal. Chromosome rearrangements will be eliminated if carriers bear negative features. If, however, their effect is superior to the ancestral genotype/karyotype, after passing the bottle neck towards homozygosity, the progeny will be positively selected. Such positive effects may be differential gene expression or advantageous linkage of distinct alleles, for instance. In the homozygous condition, positively selected chromosome rearrangements, and even selectively neutral ones, may contribute as **initial events towards speciation** (for review: [14]), because usually they act as fertility barriers.While the correct DSB repair during meiotic prophase I results in cross-overs and leads to new combinations of pre-existing alleles, mis-repair of DSBs at any developmental stage can lead to **deletion** (via end-digestion), or to sequence **insertion** (e.g., via conversion of more than the missing sequence, via transposon invasion or via alien chromatin introgression in interspecific hybrids used in crop breeding; see Figure 1) into the break. Deletions and/or insertions create a genetic novelty which is either positive, negative or selectively neutral. Positive or neutral mutations increase genetic diversity; the latter as a playground for future mutation and selection processes.If there is a (genetically fixed) bias of DSB repair towards either deletions or insertions, **shrinking or expansion of the genome** would be the corresponding long-term consequence in a population, as long as the bias is maintained (Figure 5; for review: [11]). This might explain the C-value paradox [15], which means that the genome size is not correlated with the genetic complexity of organisms.

5.In particular, genome expansion via active **retroelement amplification** and dispersion is eventually the result of DSB repair [16] biased towards insertion mediated by a retroelement-encoded integrase.6.Erroneous repair of DSBs can generate such primary chromosome rearrangements which can in turn be linked directly and/or via meiotic segregation errors with **dysploid chromosome number alteration** in both directions (Figure 6) (for review: [13]). Reciprocal translocation with breakpoints close to the centric ends of two acro- or telocentric chromosomes, which yield a large metacentric product and a small centric (or acentric) one, can reduce the chromosome number, if subsequent meiotic loss of the small product is tolerated [17] (Figure 6A). Similarly, insertion of a chromosome with breakpoints at both termini into a break within the centromere of a recipient chromosome reduces the chromosome number, if two telomeres and one centromere get lost or the recipient centromere becomes inactive (Figure 6B; [13]). If a metacentric gets broken in the centromere region in a way in which both fragments maintain centromere function and get stabilized by telomeric sequences, the chromosome number increases (Figure 6C, arrow to the left). This process can be reversible by translocation between these novel centric ends (Figure 6B, arrow to the right).

Individuals, heterozygous for two translocations with breakpoints near centromeres in one metacentric and two acrocentrics, may mis-segregate during meiosis I. Instead of correctly balanced segregation of the translocation chromosomes, four acrocentrics may segregate into one, and the corresponding two metacentrics into the other daughter nucleus. This mis-segregation simultaneously changes the haploid chromosome number by +1 or −1 compared to the ancestral situation, and was proven experimentally (Figure 6D; for review: [13] with references for experimental evidence in the plant *Vicia faba*).

7.In addition to primary rearrangements (deletion, inversion, translocation), **secondary rearrangements** (Figure 7) also depend on DSBs. Secondary rearrangements may occur in individuals which are heterozygous for two rearrangements with one chromosome involved in both rearrangements. If meiotic cross-over takes place between partially homozygous regions of rearranged chromosomes (flanked by different regions on either side of the cross-over), a newly rearranged chromosome segregates to one daughter nucleus and the re-established wild-type chromosome to the other. This pathway was also experimentally proven for plants, and might occur in other eukaryotes as well (for review: [13]).

8.Programmed DSBs take place during **V(D)J-recombination of immunoglobulin genes** in the adaptive immune system of vertebrates (for review: [18,19]). Immunoglobulins are the antibodies which recognize and neutralize antigenic proteins of invading pathogens, thus mediating disease resistance. In case of pathologic overreaction of the immune system, antibodies can cause allergies, when directed against harmless environmental antigens, or autoimmune diseases when directed against the body’s own proteins.9.DSBs, mediated by ‘domesticated’ transposases, play an essential role in **chromatin elimination.** Chromatin elimination (or diminution) occurs, e.g., in protozoans, where the chromosomes of generative micronucleus are fragmented into many, much smaller (sometimes gene-sized) chromosomes of the vegetative macronucleus, removing the interspersed repetitive sequences (for review: [20]), or in somatic stem cells of roundworms (e.g., [21]). Exceptionally, B chromosomes can be eliminated from plant organs [22].10.**Programmed cell death** (apoptosis) is another phenomenon accompanied by endonuclease-mediated DSBs, degrading nuclear DNA into small pieces (for review: [23]). Apoptosis represents a developmentally or extrinsically triggered suicidal cell destruction.11.Cancerogenesis of several tissues is also considered to start with multiple chromosome breaks as a consequence of a sudden genotoxic stress in a single cell. Such an event of catastrophic accumulation of DSBs (chromosome pulverization) and subsequent mis-repair leads simultaneously to dozens to hundreds of chromosomes rearrangements (Figure 8). The phenomenon is called **chromothripsis** [24]. The derivatives of the affected cell will mostly die (bottle neck) until viable versions (the malign cells) with the ability of rapid propagation prevail. This process is called evolution by several researchers (for review: [14]). True cancerogenesis is not known for plants. Nevertheless, multiple breakages and rearrangements occur during plant evolution (e.g., [25]). However, we cannot be sure whether evolutionarily fixed events appeared in a single cell, or subsequently over generations.

## 6. Targeted DSBs Can Modify Genetic Information for Research, Breeding and Gene Therapy

Recent developments (in particular variants of the CRISPR/Cas technology) allow for precise artificial DSB targeting. The DSBs can be generated by Cas nucleases which are guided by an RNA complementary to the target site, thus enabling site specific mutagenesis, usually via small deletions or insertion that inactivate a gene of interest. Recently, genes can be edited by precise base edition without induction of DSBs [26]. Alternatively, when additional (partially) homologous donor sequences are supplied which become integrated into the break via HDR, gene replacement with a desirable allele, or introduction of novel genetic information can be achieved (for review: [27,28,29,30,31]). Both approaches are useful for basic research (testing and modification of gene functions) and for breeding purposes (to prevent negative or promote positive traits in crops and livestock). Therapy of gene-mediated human diseases is also envisaged [26]. Finally, there are attempts to generate by targeting DSBs at distinct loci meiotic cross-overs [32], specific chromosome rearrangements to recapitulate or reverse evolutionary rearrangements [33,34], or de novo rearrangements (e.g., [35]) which might initiate speciation, as well as to domesticate de novo, e.g., orphan crops (for review [36]). 

## 7. Concluding Remarks

Although DSBs represent a severe risk for stable inheritance of genetic information, and may initiate cancerogenesis, a series of mechanisms emerged during evolution to overcome this risk by re-establishing the original status. Moreover, various routes of DSB induction and/or (mis-)repair not only contribute to stable inheritance and differentiation (chromatin elimination), but even to adapt to environmental challenges (e.g., adaptive immunity) and to promote breeding, gene therapy and speciation. Finally, despite the ambivalent nature of DSBs, evolutionary aspects might eventually turn the balance more to advantages than disadvantages.

## Figures and Tables

**Figure 1 ijms-22-05171-f001:**
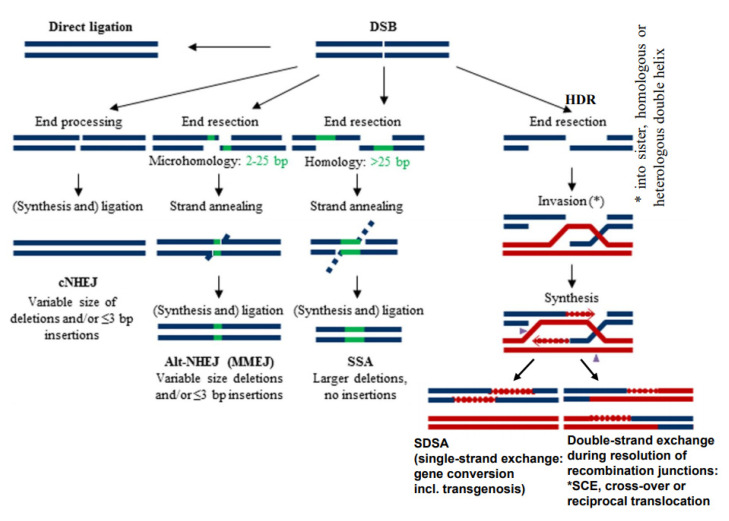
Different modes of DSB repair (modified according to [11])**.** Various degrees of end resection, homology requirement and mutation outcome distinguish conservative non-homologous end-joining (cNHEJ), alternative or microhomology-dependent end-joining (altNHEJ/MMEJ) and single-strand annealing (SSA). All three differ from homology-dependent repair (HDR), where, after end resection, single-stranded overhangs invade an undamaged (partially) homologous double helix for synthesis over the gap region prior to re-annealing and ligation (synthesis-dependent strand annealing = SDSA). The latter results in gene conversion. In the same way, transgenic donor DNA can become integrated. Alternatively, the recombination structures (Holiday junction(s)) can be resolved in a way resulting in double-strand exchange between the damaged and the template molecule. Depending on the template molecule, double-strand exchange leads to sister chromatid exchange (SCE), meiotic cross-over or different types of rearrangements (see Figure 2).

**Figure 2 ijms-22-05171-f002:**
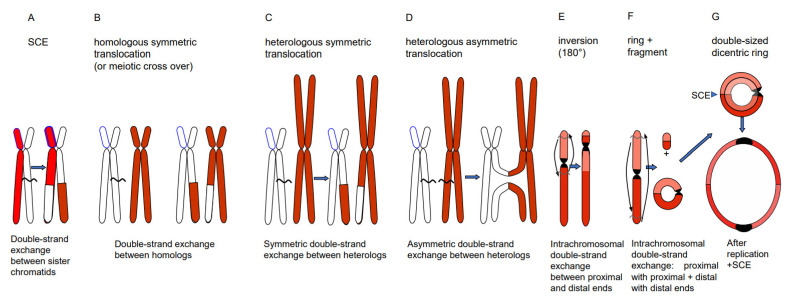
Different types of double-strand exchange. Via HDR (or NHEJ between two DSBs) the outcome of double-strand exchanges can be a SCE (**A**), a meiotic cross-over or a homologous translocation (**B**), a symmetric (**C**) or asymmetric (**D**) heterologous reciprocal translocation, an inversion (**E**) or a ring chromosome plus a fragment (**F**). A centric ring chromosome results after replication and odd numbers of SCEs in double-sized dicentric rings (**G**), which are unstable, due to bridge formation, and disrupted during nuclear division (see Figure 3).

**Figure 3 ijms-22-05171-f003:**
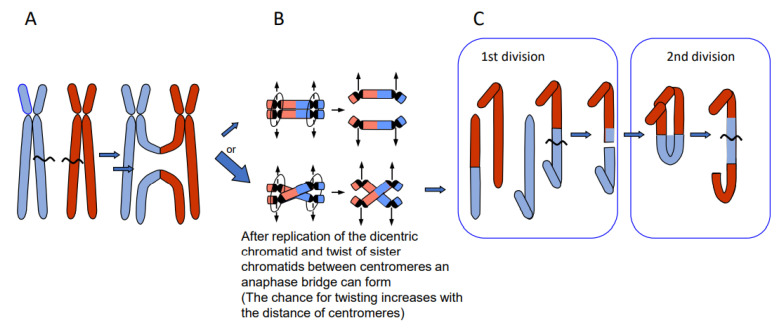
Breakage-fusion-bridge cycles of dicentric products from asymmetric reciprocal translocation (modified according to [13])**.** If the dicentric product of an asymmetric translocation (**A**) is replicated and the sister chromatids undergo a twist between the two centromeres (**B**, lower panel), a mitotic bridge will occur and break during first division (**C**, left) leading to a duplication (upper product) and a corresponding deletion (lower product). Fusion of the break ends between sister chromatids (after replication) leads to complex rearrangements in the 2nd division (shown for the upper product of the 1st division).

**Figure 4 ijms-22-05171-f004:**
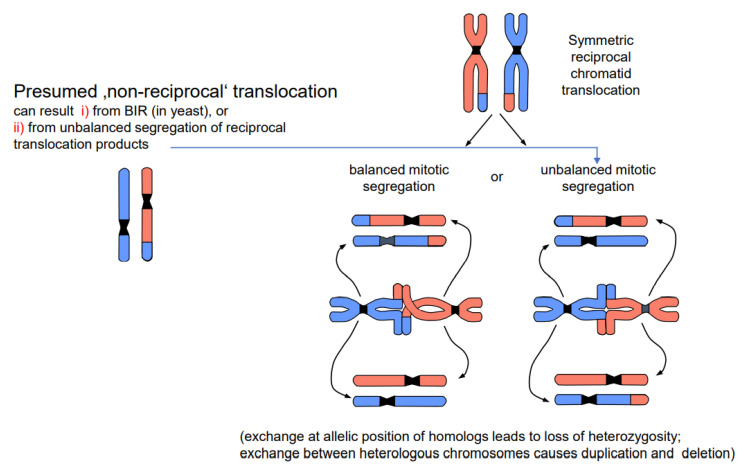
Balanced vs. unbalanced segregation of symmetric reciprocal translocation products (modified according to [13])**.** A chromosome constellation presumed to represent a non-reciprocal translocation is shown on the left; the most likely interpretation is shown on the right. While balanced segregation separates translocated and wild-type chromatids into different daughter nuclei, unbalanced segregation (one translocated and one wild-type chromatid in each daughter nucleus) gives the impression of non-reciprocity. Unbalanced segregation leads to partial loss of heterozygosity in the case of equal translocation between homologous chromosomes, and to duplications and deletion in the case of translocation between heterologous chromosomes.

**Figure 5 ijms-22-05171-f005:**
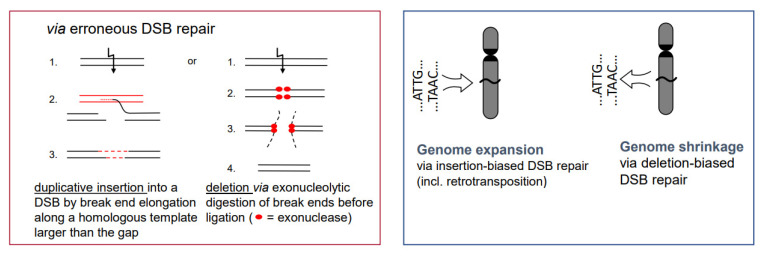
Insertion- or deletion-biased DSB repair can lead to expansion or shrinkage of genomes. The inserted sequence can also be a transposable element. Preferential dysploid chromosome number reduction can also be considered as deletion-biased DSB mis-repair (see Figure 6).

**Figure 6 ijms-22-05171-f006:**
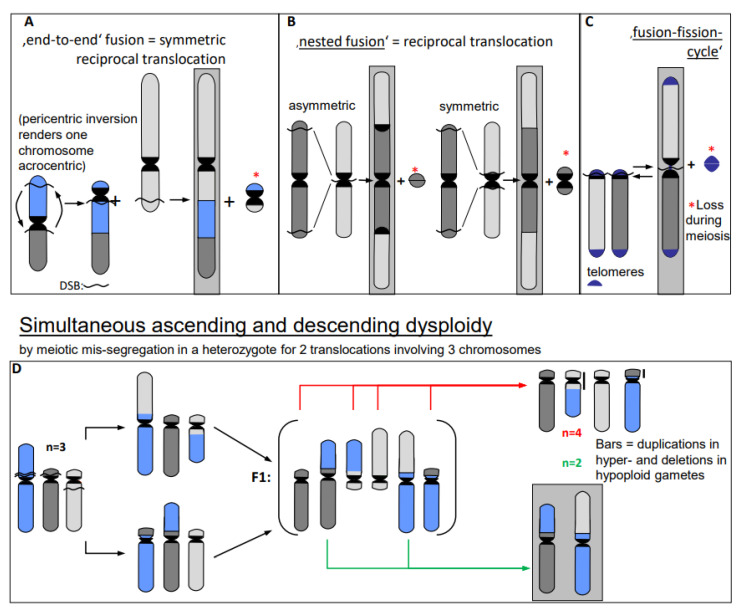
Dysploid chromosome alterations (modified according to [11])**.** After multiple breakage in (**A**–**C**), the second small translocation product* (even if containing a centromere) is frequently lost during meiosis (**A**–**C**) [17]. Meiotic mis-segregation from multivalents in organisms heterozygous for distinct translocations yield germ cells with the simultaneous increase and decrease of the parental haploid chromosome number (**D**). The increase is accompanied by small duplications and the decrease by the corresponding deletions (vertical bars in (**D**)). Chromosome configurations of descendent dysploidy are framed in gray.

**Figure 7 ijms-22-05171-f007:**
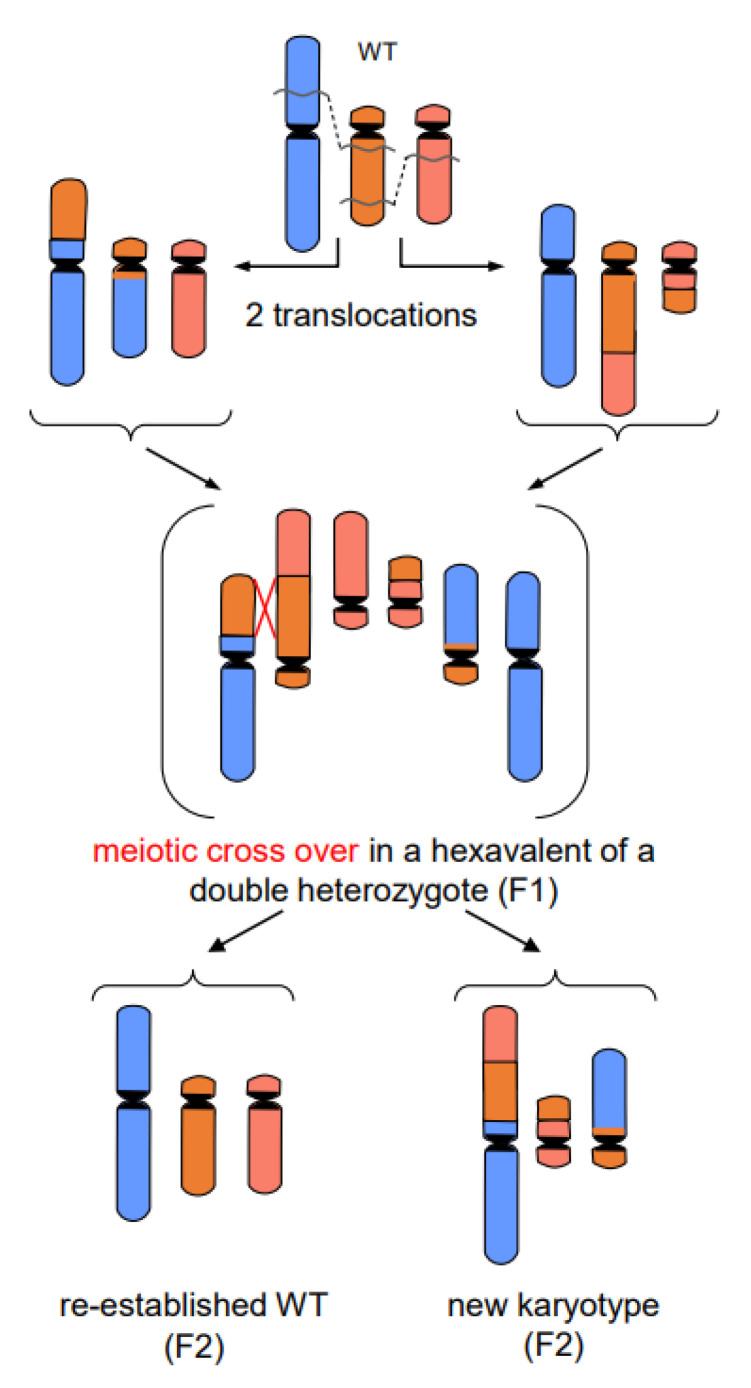
Secondary rearrangement (modified according to [13])**.** The example shows the meiotic crossing-over at partially homologous chromosome regions (flanked by non-homologous regions) of two different translocation chromosomes. Such events occur in individuals which are double-heterozygous for two translocations involving three chromosomes (one chromosome participating in both translocations). The cross-over products segregate into daughter nuclei of which one harbors the new secondarily rearranged chromosome, and the other the reconstituted wild-type chromosome complement.

**Figure 8 ijms-22-05171-f008:**
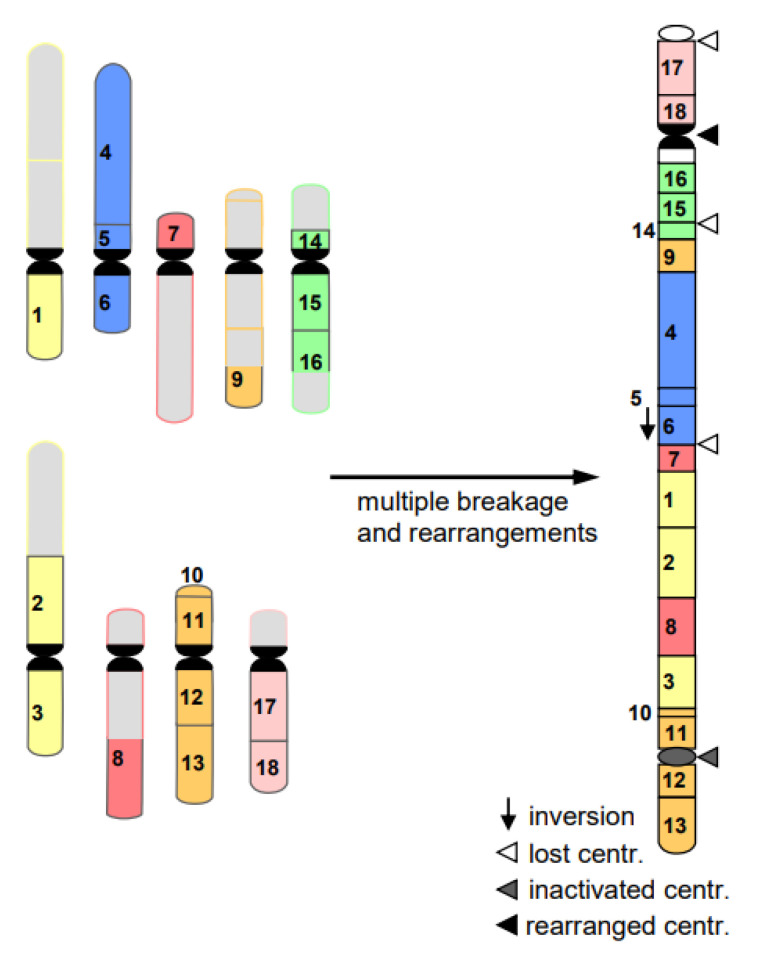
Scheme of chromothripsis. Multiple breakages in several chromosomes result, via mis-repair, in multiple rearrangements. On the right, a large rearranged chromosome is shown, consisting of several parts of nine original chromosomes (left). The potential fate of the remaining chromosomes on the left is omitted for simplicity. The majority of the resulting cells die; few may prevail as malignant ones.

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
