# Peer review of "Boon and Bane of DNA Double-Strand Breaks"

_ijms, 2021, doi:10.3390/ijms22105171_

Round 1
Reviewer 1 Report
Dear Author,
I have a few minor comments on the manuscript. The beginning section lacks information about the role of ROS in DSB formation. I also miss a short paragraph discussing the purpose of the paper. So that the reader can quickly get an idea of what to expect in the later parts of the paper. Figure 2 is too small and thus unreadable. I also have the impression that there are different fonts in the descriptions in the figures and in the case of figures 1, 2, and 3 the font should be larger. In the final part of the paper, I also miss the indication of the role of DSBs in crop breeding when developing lines containing alien chromatin introgressions.
Best regards,
Author Response
I have a few minor comments on the manuscript. The beginning section lacks information about the role of ROS in DSB formation.
Response: I mentioned ROS as DSB-inducing agent in line 28.
I also miss a short paragraph discussing the purpose of the paper. So that the reader can quickly get an idea of what to expect in the later parts of the paper.
Response: I inserted a corresponding sentence at the end of the Abstract (lines12-13).
Figure 2 is too small and thus unreadable. I also have the impression that there are different fonts in the descriptions in the figures and in the case of figures 1, 2, and 3 the font should be larger.
Response: the font size of the text within figures 1, 2 and 3 has been increased.
In the final part of the paper, I also miss the indication of the role of DSBs in crop breeding when developing lines containing alien chromatin introgressions.
Response: I mentioned chromatin introgression in crop breeding as a consequence of DSB within interspecific hybrids (line 189).
Reviewer 2 Report
The review is written clearly, briefly and can be recommended as a study guide for undergraduate and graduate students. The review considers possible types of DNA breaks and DNA repair. In the review, the author made the main emphasis on the fact that mis-repair in plants can lead to stable inheritance and facilitate adaptation to unfavorable environmental influences. This is an important conclusion. The development of methods that alter genetic information in certain places in the future can achieve great success in breeding practice.
There are some small remarks in the review:
1. in figure 2, inscriptions are poorly visible
2. in figures 4 and 8, you can remove the inscription figure 4) and figure 8)
3.references can be combined - line 287
4.reference 17 double - line 335
Author Response
- in figure 2, inscriptions are poorly visible
Response: The text within figures 1, 2 and 3 has been enlarged
- in figures 4 and 8, you can remove the inscription figure 4) and figure 8)
Response: The headings inside Figs 4 and 8 have been removed.
3.references can be combined - line 287
Done: [26-30] (line 294)
4.reference 17 double - line 335
The duplication (line 343) has been removed